# Cortical cell assemblies and their underlying connectivity: An *in silico* study

**András Ecker** [ID] [◐], **Daniela Egas Santander** [ID] [◐], **Sirio Bolaños-Puchet** [ID], **James B. Isbister** [ID], **Michael W. Reimann** [ID] *

Blue Brain Project, École polytechnique fédérale de Lausanne (EPFL), Campus Biotech, Geneva, Switzerland

◐ These authors contributed equally to this work.

* michael.reimann@epfl.ch

**Data Availability Statement:** The 2.4 mm³ subvolume of the juvenile rat somatosensory cortex (containing 211,712 neurons and their connectivity) used for the in silico experiments in this study has been deposited at Zenodo in

## Abstract

Recent developments in experimental techniques have enabled simultaneous recordings from thousands of neurons, enabling the study of functional cell assemblies. However, determining the patterns of synaptic connectivity giving rise to these assemblies remains challenging. To address this, we developed a complementary, simulation-based approach, using a detailed, large-scale cortical network model. Using a combination of established methods we detected functional cell assemblies from the stimulus-evoked spiking activity of 186,665 neurons. We studied how the structure of synaptic connectivity underlies assembly composition, quantifying the effects of thalamic innervation, recurrent connectivity, and the spatial arrangement of synapses on dendrites. We determined that these features reduce up to 30%, 22%, and 10% of the uncertainty of a neuron belonging to an assembly. The detected assemblies were activated in a stimulus-specific sequence and were grouped based on their position in the sequence. We found that the different groups were affected to different degrees by the structural features we considered. Additionally, connectivity was more predictive of assembly membership if its direction aligned with the temporal order of assembly activation, if it originated from strongly interconnected populations, and if synapses clustered on dendritic branches. In summary, reversing Hebb's postulate, we showed how cells that are wired together, fire together, quantifying how connectivity patterns interact to shape the emergence of assemblies. This includes a qualitative aspect of connectivity: not just the amount, but also the local structure matters; from the subcellular level in the form of dendritic clustering to the presence of specific network motifs.

## Author summary

It is widely known in the neuroscience field that cells that fire together wire together, forming so called "*cell assemblies*". However, on a quantitative level the picture is more complicated, because neurons can participate in several spatially distributed assemblies and receive synaptic inputs from many local and long-range sources. The effects of different connections on different assemblies are difficult to analyze because simultaneous recordings of neuronal activity and their connections are only feasible for small sets of

SONATA format and is publicly available at the following DOI: 10.5281/zenodo.7930275. The simulator front-end that loads the SONATA model, and instantiates the simulation to be run in CoreNEURON is also publicly available at GitHub or under the following DOI: 10.5281/zenodo.8075202. Assemblies from 10 simulation repetitions (with different random seeds), their consensus, and underlying significant spike times, and the whole excitatory connectivity matrix have been deposited at Zenodo and is publicly available at the following DOI: 10.5281/zenodo.8052721. Analysis code that created (and can easily open the dataset above) is publicly available at GitHub or under the following DOI: 10.5281/zenodo.8112725.

**Funding:** This study was supported by funding to the Blue Brain Project, a research center of the École polytechnique fédérale de Lausanne (EPFL), from the Swiss government's ETH Board of the Swiss Federal Institutes of Technology. The funders had no role in study design, data collection and analysis, decision to publish, or preparation of the manuscript.

**Competing interests:** The authors have declared that no competing interests exist.

neurons in a given region. Thus, we turned to a complimentary, simulation-based approach using a large-scale cortical microcircuit model with non-random, biorealistic connectivity. We found different types of assemblies that differed in how much they were determined by connections from local or long-range sources. Additionally, we characterized ways in which connectivity can tie a neuron into an assembly more efficiently, for example through dendritic clustering of synapses.

## Introduction

The formulation of the cell assemblies concept goes back to Hebb 1949 [1], who defined them loosely as "*a diffuse structure comprising cells in the cortex*". In the past 70 years, the sequential activation of groups of neurons, the Hebbian "*phase sequence*" was linked to several complex cognitive processes, reviewed in Refs [2, 3]. Hebb's idea was later paraphrased as "*cells that fire together, wire together*" [4], giving it both a structural, and a functional side. In this article we will concentrate on quantifying how the cortical structure underlies its neurons' co-firing function, but linking these groups of co-active neurons to cognitive processes is outside of our scope.

Cell assembly research rejuvenated in the hippocampus field when spikes could be reliably sorted from recordings with tetrodes and therefore neurons could be grouped in co-firing ensembles [5–8]. The introduction of modern *in vivo* two-photon calcium imaging into the field, with its improved scalability and stability over time, allowed Bathellier et al. [9] and Carrillo-Reid et al. [10] to detect cell assemblies in auditory and visual cortices, where they showed how even a small set of them can serve as a backbone for cortical coding. These, and studies that followed [11–13] contributed greatly to our understanding of the functional role of the Hebbian cell assemblies. For example, they showed that stimulus-evoked cell assemblies can also be spontaneously active, demonstrating that they are at least partially determined by local circuitry. On the other hand, they could not make claims about the specific patterns of synaptic connectivity they originate from, as they could only predict functional connectivity from correlations in neuronal activity, but did not have access to the underlying structural connectivity of the neurons recorded. Additionally, results based on calcium imaging are limited to the superficial layers of the cortex, missing potential assemblies in the deeper layers, which would be of great interest as they serve as the output of the cortex [14, 15].

Early theoretical work in the field explored the potential link between memories and cells that fire and therefore wire together, concentrating on the storage and retrieval of memories in strongly recurrent networks, such as the CA3 area of the hippocampus [16]. Theories evolved and improved, but modeling studies about cell assemblies still concentrate on plasticity rules underlying the learning, storage and recall of various patterns [17–22]. Thus, their focus lies on how function shapes structure, with little or no emphasis on the biologically accurate aspects of structural connectivity, such as low connection probabilities and an abundance of directed motifs [23–25].

On the other hand, the perspective can be reversed: how does a more bio-realistic structural connectivity influence a neuron's membership in one or more assemblies? Or on a more general level: how does structure determine function? Additionally, how does innervation from different sources, such as local connectivity and various thalamic afferents, interact to shape assembly membership? Finally, are the afferent synapses from fellow assembly neurons scattered across the dendritic tree, or clustered on single branches, employing the nonlinear computational capabilities of dendrites [26–28]?

In order to provide insights into these questions, we employed an *in silico* approach, using an improved version of the detailed, large-scale (somatosensory) cortical circuit model of Markram et al. [29] (Fig 1A), simulating the activity of over 200 thousand neurons in response to a stream of thalamic input patterns. In the model, we have access not only to the spiking activity of every neuron, but also to the entire connectome, including dendritic locations of synapses. We then considered the established, purely functional definition of cell assemblies as neurons that fire together more than expected in shuffled controls. Therefore, functional assemblies across all cortical layers were detected using a combination of previously published methods [10, 30] (Fig 1B). These methods stem from detecting significant peaks in firing rates, which in living tissue occur even in the absence of controlled stimuli, therefore a lot of interest is devoted to spontaneous assemblies *in vivo* [10, 11, 13] and "neural avalanches" *in vitro* [31]. On the other hand, we focused our simulation setup on matching the evoked activity state of the cortex [32] with reduced background activity and therefore analyzed evoked assemblies only. We then analyzed their underlying structural connectivity, searching for rules that could explain assembly membership (Fig 1C and 1D). As we expected these rules to differ between excitatory and inhibitory connections, we focused our analyses on the excitatory subnetwork only. This analysis of the structure-function relation could be readily applied to assemblies detected with other methods [8, 33, 34].

We found that the structural strengths of afferents from various sources explained significant portions of the uncertainty of a neuron's membership in an assembly (Fig 1D). Specifically, innervation from VPM (ventral posteriomedial nucleus of the thalamus) explained up to 30%, and up to 10% from POm (posteriomedial nucleus of the thalamus). Strength of innervation through recurrent local connectivity explained up to 22%. The relative magnitudes of these effects differed between assemblies, depending on the order of their activation: The later an assembly was active during a stimulus, the more it was determined by the structure of recurrent connectivity and comparatively less by thalamic innervation. Additionally, the effect of innervation strength on assembly membership can be much larger if the innervating

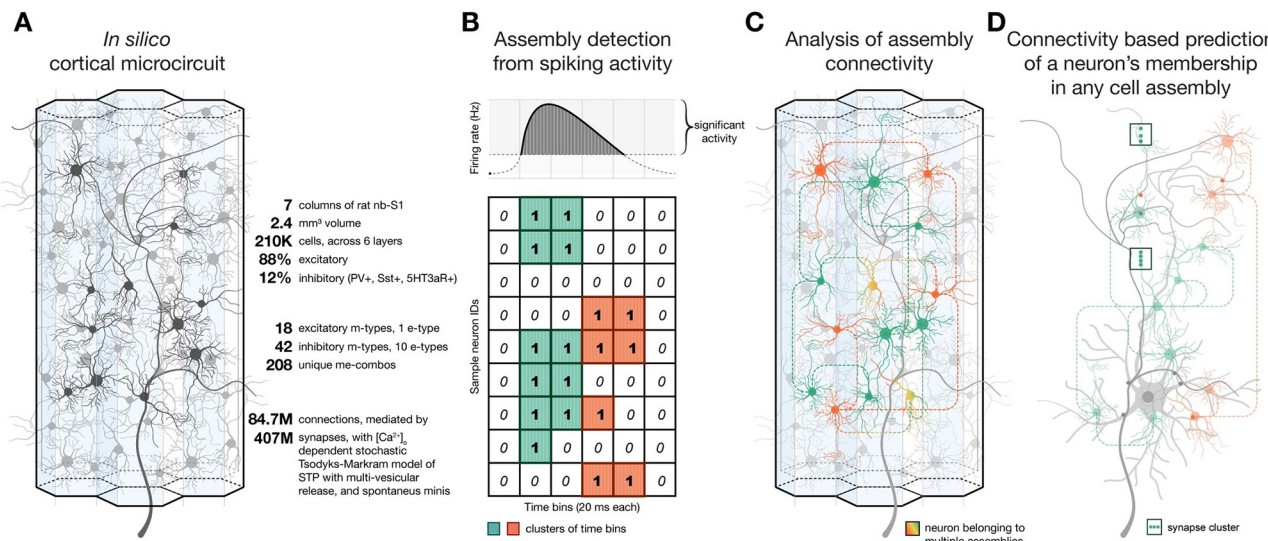

**Fig 1. Pipeline summary. A**: Schematics and quick facts about the detailed, large-scale cortical microcircuit that was used several times before to study the relationship of cortical structure and function [25, 35, 36]. **B**: Schematics of the assembly detection pipeline from the spiking activity of 186,665 excitatory neurons in the circuit. **C**: Analysis of the connectivity of cell assembly neurons. **D**: Derivation of assembly membership probability based on different features of structural connectivity.

population is highly interconnected within itself, and if the synapses are tightly clustered on the targeted dendrites (Fig 1D). The highly non-random structure of this connectome provides a more efficient wiring; implementing ensembles of reliably co-firing neurons with fewer synaptic connections than expected by chance.

## Results

### A diverse set of assemblies can be detected from network simulations

We simulated the electrical activity of a model of 2.4 $mm^3$ of cortical tissue, comprising 211,712 neurons in all cortical layers in an *in vivo*-like state. The model is a version of Markram et al. [29] with anatomical improvements outlined in Reimann et al. [37] and physiological ones in Isbister et al. [32] (Fig 1A). We consider the activity to be *in vivo*-like, based on a comparison of the ratios of spontaneous firing rates of sub-populations, and responses to brief thalamic inputs to *in vivo* results from Reyes-Puerta et al. [38] (as described in Isbister et al. [32]). A stream of thalamic input patterns was applied to the model (see Methods), and the neuronal responses recorded (Fig 2a1). The circuit reliably responded to the brief stimuli with a transient increase in firing rate. This led to a slight shift to the right of the tail of the firing

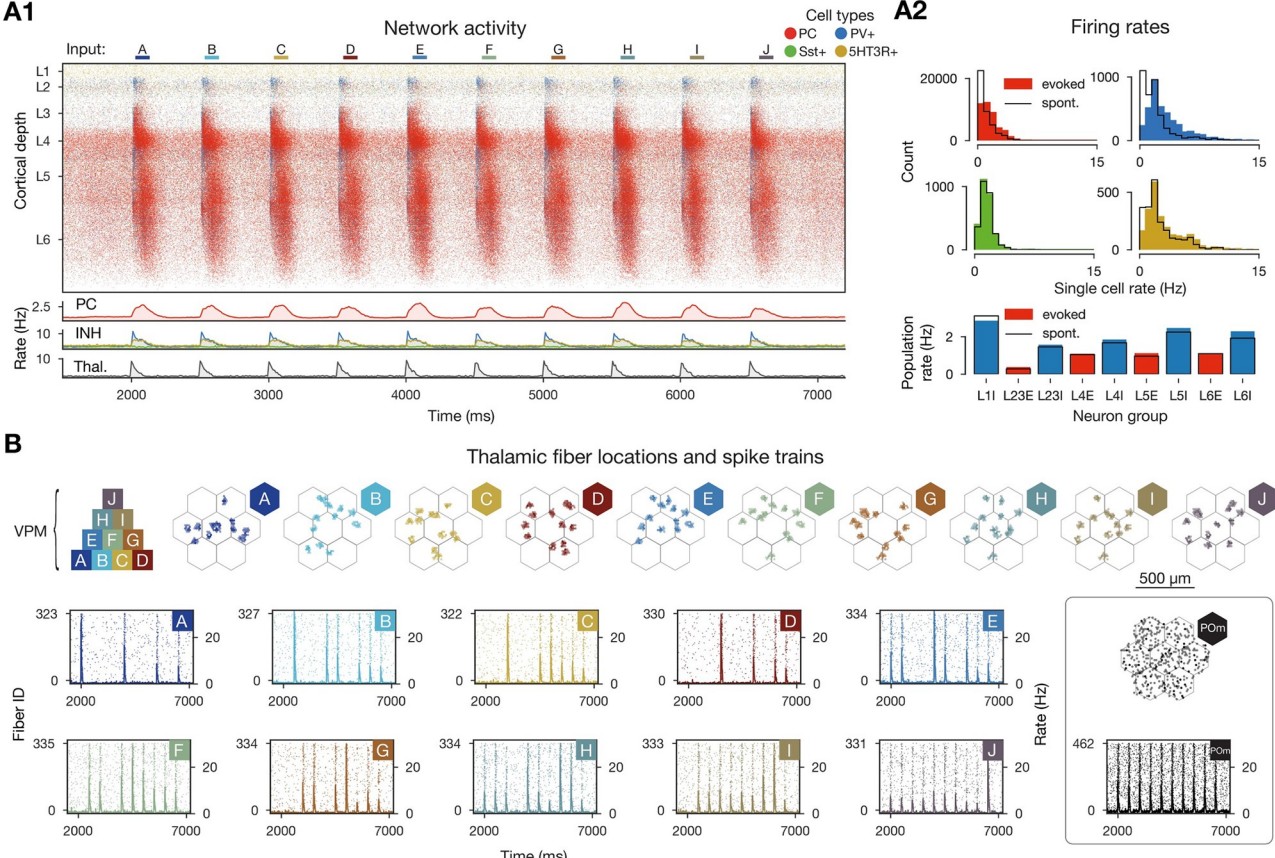

**Fig 2. *In vivo*-like activity *in silico*. A1**: Raster plot of the microcircuit's activity with 628,620 spikes from 98,059 individual neurons and the population firing rates below. The y-axis shows cortical depth. (As cortical layers do not have the same cell density, the visually densest layer is not necessarily the most active—see a2 bottom.) **A2**: Single cell firing rates (in excitatory and 3 classes of inhibitory cells) and layer-wise inhibitory and excitatory population firing rates in evoked (showed in a1) and spontaneous (not shown) activity. **B**: Top: pyramid-like overlap setup of VPM patterns, then the centers of the VPM fibers in flat map space. Bottom: raster plots of VPM fibers forming each of the patterns for the stimulus stream in a1 (i.e., from pattern A at 2000 ms to pattern J at 6500 ms). On the right: same for non-specific (POm) input.

rate distribution from the spontaneous state (Fig 2A2), in line with experiments [39]. Stimuli were a combination of simultaneous inputs from thalamic nuclei VPM and POm. Inputs from POm were non-specific, with the same fibers being active each time. Inputs from VPM were repeated presentations, in random order, of 10 different input patterns [35], with varying degrees of overlap (Fig 2B): 4 base patterns with no overlap (A, B, C, D), 3 patterns as combinations of two of the base ones (E, F, G), 2 patterns as combinations of three of the base ones (H, I), and 1 pattern as a combination of all four base ones (J). The overlap of these patterns can also be seen through the raster plots of their corresponding VPM fibers (Fig 2B bottom). For example the fibers corresponding to pattern A peak when stimulus A is presented, but also with 50% of the amplitude when E is presented.

Using a combination of the algorithms of Carrillo-Reid et al. [10] and Herzog et al. [30], we detected functional assemblies in 125-second-long recordings of simulated neuron activity while receiving the random input stream (25 repetitions of all 10 patterns with 500 ms inter-stimulus interval, see Methods). Briefly, the assembly detection algorithm first groups neuronal activity into 20 ms time bins, and identifies those with significantly increased firing rates [7, 10] (see Methods, Fig 3A). Then, these time bins are hierarchically clustered based on the cosine similarity of their activation vector, i.e., the vector of the number of spikes fired in the time bin for each neuron [12, 13] (S1A Fig, Fig 3B1). The threshold for cutting the clustering tree into clusters is determined by minimizing the resulting Davies-Bouldin index [40] (see Methods, S1 and S2 Figs for lower dimensional representations). Finally, these clusters correspond to the functional assemblies, with a neuron being considered a member if its spiking activity correlates with the activity of an assembly significantly more strongly than chance level [12, 30]. This means that in each time bin only a single assembly is considered active, but neurons can be part of several assemblies (see Methods, Figs 1B and 3B2).

We found that assemblies were activated in all stimulus repetitions, a series of two to three assemblies remained active for 110 ± 30 ms and the number of neurons active in consecutive time bins follow a Gaussian distribution (Fig 3B2, S1D Fig). Activation probability and duration depended on stimulus identity; the stimulus associated with the strongest response elicited 1.6 times as many significant time bins than the stimulus associated with the weakest response. Not only were individual assemblies associated with only a subset of stimuli, but they also had a well-preserved temporal order, with some of them appearing early during a stimulus, and others later. Based on this, from now on we will refer to them as *early, middle*, and *late assemblies*, and will order them in the figures accordingly.

An assembly comprised on average 7 ± 3.1% of the simulated excitatory neurons, with late assemblies being about 3.7 times larger than early ones (Fig 3C). Note in particular that since assemblies are stimulus specific and we present only 10 input patterns, a plurality (in fact 40%) of neurons do not belong to any assembly. In terms of spatial distribution of assembly neurons, the early assemblies appear more "*patchy*" (small distinguishable clusters), and by visual inspection can be mapped back to the locations of VPM fibers corresponding to the stimuli that activated them (Figs 2B and 3C top). Moreover, their layer profile mimics that of VPM fiber innervation [37, 41] (S4A Fig), indicating that these assemblies may be determined by direct thalamic innervation. On the other hand, the late assembly neurons are more evenly distributed, and cover the entire surface of the simulated circuit, even beyond the range of VPM fiber centers, and are found mostly in deeper layers of the cortex. Middle assembly neurons are somewhat in-between, both in spatial distribution and depth profile. Although we found late assemblies to be nonspecific (at the chosen clustering threshold), early and middle assemblies belonging to the same stimulus occupy similar regions in space and can be shown to have a relatively high (25%) overlap of neurons (Fig 3D left). This also means that single neurons belong to several assemblies (up to 7 out of 11, Fig 3D right). In conclusion, the time course of

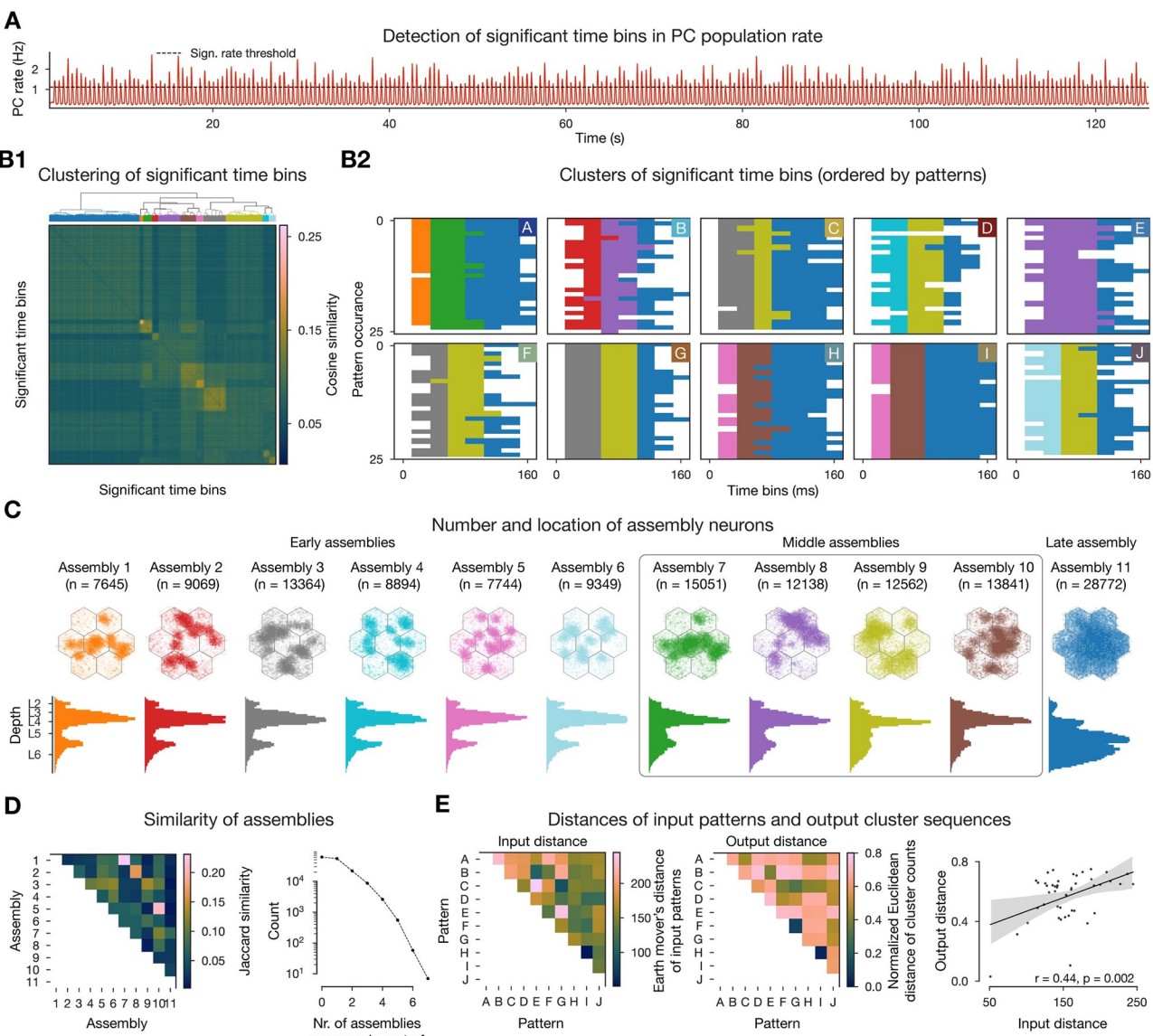

**Fig 3. Cell assembly detection. A**: Population firing rate of excitatory neurons with the determined significance threshold. **B1**: Hierarchical clustering of the cosine similarity matrix of activation vectors of significant time bins (above threshold in A, see Methods). **B2**: Clustered significant time bins ordered by patterns presented. Colors indicate assembly identity and will be used systematically throughout the remainder of the manuscript. **C**: Number and location of neurons in each cell assembly: flat map view on top, depth-profile below. **D**: Jaccard similarity of cell assemblies and number of neurons participating in different number of assemblies. **E**: Input-output map: Input distance is calculated as the Earth mover's distance of the VPM fiber locations, this metric is small when the activated fibers have large intersections and large otherwise (see Fig 2B and Methods). The output distance is the normalized Euclidean distance of pattern evoked time bin cluster counts (counts of different colors in the matrices above in B2, Methods) this measure is minimized if the same assemblies are active at the same time.

stimulus responses are well-preserved and reliable enough (although with some temporal jit-ter) to be simplified into a sequence of distinct sets of functional cell assemblies.

Although stimulus-specific, the time courses are not unique (see the responses to patterns H and I in Fig 3B2). Thus, we wondered to what degree is the overlap of assemblies associated with the overlap of the input patterns given as stimuli? When we compared the distances between the locations of the VPM fibers making up an input pattern, and the assembly

sequences detected from the network activity, we found a significant linear trend, i.e., patterns that are close in the input space (e.g. H and I) are close in the "*output space*" defined as the counts of individual assemblies popping up for a given stimuli across repetitions ([Fig 3E]). Thus, the activation sequence of cell assemblies can be seen as a low-dimensional representation of the complex, high-dimensional activity of the circuit's response to different stimuli. The data points are highly variable for mid-sized input distances, and the linear trend gets weaker with increasing number of assemblies ([S2C Fig]). In summary, increasing the number of assemblies by cutting the clustering tree differently improves the separation of inputs at the cost of reducing the correlation between input and output distance. As our aim was not to build an ideal decoder of input patterns, in the following steps we analyze the assemblies resulting from a clustering that minimized the Davis-Bouldin index ([S2A2 Fig]).

**Assemblies detected only in the superficial layers, are reminiscent of those detected across all layers of the cortex.** Due to the limitations of traditional two-photon microscopy, *in vivo* experiments in the cortex can only detect cell assemblies in layer 2/3 (L2/3) [9–13], whereas *in silico* we detect assemblies across all layers. This begs the question: are *in silico* assemblies different than the ones detected *in vivo*, or do they just cover more depth but contain similar L2/3 ones? To answer this question, we used the same data and methodology as before, but restricted our analysis to L2/3 neurons only ([S3 Fig]). When detecting assemblies exclusively from the spiking activity of L2/3 pyramidal cells, we got the same results overall, but with some specific differences. First, we found significant time bins for a shorter range (up to 100 ms, which corresponds to our definition of early and middle assembly time windows). And second, there was a stronger one-to-one correspondence between stimuli and assemblies, e.g. stimuli H and I correspond to different early assemblies in L2/3 (compare [Fig 3B2] and [S3A2 Fig]). We compared the L2/3 assemblies to the original full assemblies and found that the early ones can get mapped to the L2/3 ones relatively well, in terms of the Jaccard similarity of their respective intersections with L2/3 neurons, and by visual inspection of the spatial locations of assembly neurons ([S3D Fig]). Thus, we predict that assemblies detected *in vivo* are the superficial subset of full assemblies, and may not include late assemblies with only a small fraction of neurons in superficial layers.

## Functional assemblies are determined by structural features

It appears as if the spatial structure of the thalamic input stimuli strongly determines assembly membership. At the same time, neuronal assemblies are thought to be strongly recurrently connected [23, 24].

We generally observe that some features of structural connectivity can be predictive of the probability of assembly membership. Here, we consider features related to thalamic innervation, recurrent connectivity and synaptic clustering. We quantify the strength of this effect by means of a thresholded and signed version of the mutual information which we call their *normalized mutual information* and denote by $nI$ (for details, see [Methods]). As for mutual information, its absolute value does not require any assumptions about the shape of the dependency (e.g., linear, monotonic, etc.) between the structural feature and the probability of assembly membership. However, $nI$ deviates from regular mutual information in the following ways: First, it is positive if the probability of assembly membership increases as the value of the structural feature increases, and is negative otherwise. This allows us to show which measures promote or dissuade assembly membership. Second, its absolute value is one if assembly membership can be completely predicted from the structural feature and it is set to zero if the mutual information is not statistically significant. This allows easy assessment of significance and comparison between plots.

**Thalamic innervation explains early and middle assemblies.** We began by considering the effect of direct thalamic innervation. Having confirmed that strong direct thalamic innervation facilitated assembly membership (Fig 4A, left), we formulated the following hypothesis: pairs of neurons are more likely to belong to the same assembly if they are innervated by overlapping sets of thalamic fibers. To test this, we first consider the common thalamic indegree of a pair of neurons, i.e., the number of thalamic fibers innervating both of them. Then, for each neuron, we use its *mean common thalamic indegree* over all cells in assembly $A_n$, as the structural feature to predict its membership in $A_n$. We performed this analysis separately for innervation from the VPM and POm nuclei.

In both cases, mean common thalamic indegree with an assembly increased the probability that a neuron is part of it (Fig 4A, second, olive curve for common thalamic indegree with the same assembly). The *nI* of the mean common thalamic indegree and membership in the same assembly was on average 0.165 for VPM and 0.054 for POm (Fig 4A, right, entries along the diagonals). More specifically, 0.157 and 0.034 for early assemblies, 0.198 and 0.087 for middle, and 0.082 and 0.037 for the late assembly. In addition, cross-assembly interactions were also observed, albeit at lower levels (Fig 4A, right, off-diagonal entries). The effect was strongest for pairs of early and middle assemblies that responded to the same stimuli, e.g. assemblies 1, and 2, responding to pattern A.

The lower *nI* values for the late assembly are expected, as it contains many neurons in layers not directly innervated by thalamus (Fig 3C, S4A Fig), and its activity is largely restricted to time bins in which the thalamic input is only weakly active (80–140 ms after onset; Figs 3B2 vs. 2A1). Interestingly, common innervation with POm has the highest *nI* for middle assemblies, which seems related to the prevalence of L5 neurons in them (2.3 times more L5 cells, than in the early ones). POm targets the upper part of L5, and more importantly L1 [37, 41] (S4A Fig), where the apical tuft dendrites of thick-tufted L5 pyramidal cells reside [15, 42]. The delay caused by the long synapse to soma path distances (S4B Fig) may explains the importance of common POm innervation 40–60 ms after stimulus onset.

Having confirmed that common innervation by thalamic fibers links pairs of neurons to the same assemblies, we then considered how much more assembly membership is determined by the identity of the specific patterns used. We hypothesized that direct innervation by fibers used in a pattern increases membership probability in assemblies associated with the same pattern. Specifically, we used as a structural feature the *pattern indegree*, i.e., the total indegree of a neuron from VPM fibers used in each of the patterns.

As predicted from the previous results, probability of assembly membership grew rapidly with pattern indegree for early assemblies associated with the same pattern (Fig 4B, left). Every pattern had one early assembly strongly associated with it (except for pattern E, which only had middle and late assemblies Fig 3B2). On average, the *nI* of pattern indegree and assembly membership reached 0.26 (Fig 4B, right). A similar trend was observed for middle assemblies (mean: 0.215), while the late assembly was again an exception, with no value above 0.04, for reasons outlined above (Fig 4B third and fourth). In total, taking the stimulus patterns into account gives a 33% higher *nI* over the less specific mean common VPM innervation.

**Recurrent connectivity explains late assemblies.** Even with perfect knowledge of thalamic innervation and pattern identity the *nI* did not exceed values of 0.3, leading to the question: What other factors determine the rest? The most commonly accepted structural correlate of cell assemblies is the overexpression of recurrent connectivity motifs between participating neurons [2, 3, 23, 24]. One particular class of motifs that has been linked to neuronal function are directed simplices of dimension $k$ ($k$-simplices [25]). A $k$-simplex is a motif on $k + 1$ neurons, which are all-to-all connected in a feed-forward fashion (Fig 4E left, inset); in particular 1-simplices are directed edges and 0-simplices are single cells. Indeed, we found a strong

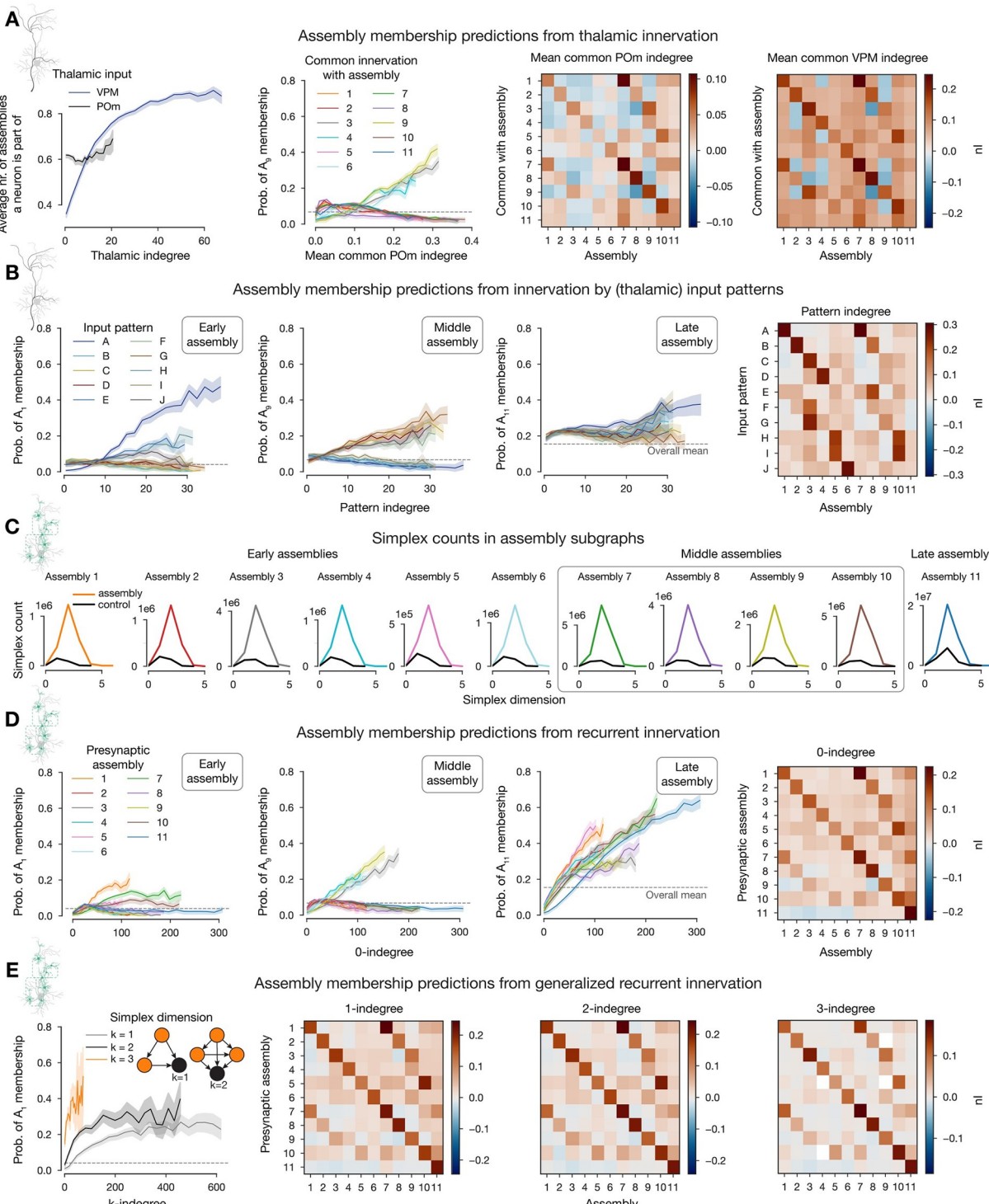

**Fig 4. Connectivity determines cell assembly membership. A-B**: Thalamic connectivity, **C-D**: recurrent connectivity (indicated with schematics from Fig 1D). **A**: First: Effect of thalamic innervation (from VPM and POm nuclei) on participation in cell assemblies. Solid lines indicate the mean and the shaded areas indicate 95% confidence intervals. Second: probability of membership in an exemplary middle assembly against mean common POm indegrees with respect to all assemblies. Third and fourth: *nI* (normalized mutual information, see Methods) of mean common thalamic indegree and assembly membership. **B**: Probability of membership in exemplary early (first), middle (second), and late (third) assemblies against pattern indegree with respect to all patterns. Fourth: *nI* of pattern indegree and assembly membership. **C**: Simplex counts within assemblies and random controls (same number of neurons with the same cell type distribution), where a simplex of dimension *k* is a motif of *k* + 1 neurons fully connected in feed-forward fashion. **D**: Probability of membership in exemplary early (first), middle (second)

and late (third) assemblies against 0-indegree with respect to all assemblies. Fourth: $nI$ of 0-indegree and assembly membership. **E**: First: Probability of membership in an exemplary early assembly against $k$-indegree with respect to the same assembly. Inset: $k$-indegree is given by the number of $k$-simplices within an assembly (orange nodes) completely innervating a given neuron (black). $nI$ of $k$-indegree and assembly membership for $k = 1$ (second), $k = 2$ (third) and $k = 3$ (fourth).

overexpression of directed simplices in the connectivity submatrices of cells within an assembly. In particular, the maximal simplex dimension found in assembly subgraphs is at least one higher than in random subgraphs generated by sampling the same number of neurons with the same distribution of cell types. Moreover, the peak of simplex counts in assembly graphs is in general one order of magnitude above these controls (Fig 4C).

Based on this we define the *k-indegree with respect to an assembly* of a neuron *i*, as the number of $k$-simplices in the assembly such that all the cells in the simplex innervate *i* (see Fig 4E left, inset). For the case $k = 0$, i.e., the number of cells in the assembly innervating *i* (the general notion of indegree), we found that it is a good predictor of membership in the same assembly, with an average $nI$ value of 0.143 (Fig 4D). This time, late assembly membership could be predicted the best, with an $nI$ of 0.224, compared to an average of 0.118 and 0.16 for early and middle assemblies, respectively (Fig 4D). Additionally, probability of late assembly membership also increased with 1-indegree with respect to all other assemblies. Conversely, 0-indegree with respect to the late assembly decreases the membership probability for early, and most of the middle assemblies (blue curves with negative slope on the first and second panels of Fig 4D). This reflects the temporal order of their activation and the fact that neurons in the deeper layers, that dominate the late assembly, mostly project outside the cortex and not back to superficial layers [14, 15]. The $nI$ values across the diagonals for $k = 1, 2$ are larger than for $k = 0$, for all assemblies except the late one. More precisely, we found $nI$ averages of 0.152 and 0.161 for early, and 0.18 and 0.188 for middle assemblies for $k = 1$ and $k = 2$ respectively (Fig 4E second and third). On the other hand, off-diagonal $nI$ values drop for increasing $k$. This shows that not only the size of the presynaptic population has an effect on the activity of a neuron, but also the connectivity patterns between them. However, the effect of these non-local interactions are stronger within an assembly than across. For $k = 3$ the $nI$ values drop, which can be explained by the narrower range of values the 3-indegree takes (Fig 4E first and fourth).

**Synaptic clustering explains late assemblies.** So far, we have only considered features which can be extracted from the connectivity matrix of the system. However, our model also offers subcellular resolution, specifically, the dendritic locations of all synapses [29, 43], which we have demonstrated to be crucial for recreating accurate post-synaptic potentials [44, 45], and long-term-plasticity [46]. This allowed us to explore the impact of co-firing neurons potentially sending synapses to the same dendritic branch, i.e., forming synapse clusters [28, 47–49]. We hypothesized that innervation from an assembly is more effective at facilitating membership in an assembly if it targets nearby dendritic locations. We therefore defined the synaptic clustering coefficient (*SCC*) with respect to an assembly $A_n$, based on the path distances between synapses from $A_n$ on a given neuron (see Methods, S5 Fig). The *SCC* is a parameter-free feature, centered at zero. It is positive for intersynaptic distances that are lower than expected (indicating clustering) and negative otherwise (indicating avoidance).

Overall, we found similar trends as for the recurrent connectivity, although with a lower impact on assembly membership. Late assembly membership was explained the best by the *SCC* with 0.114 $nI$, while early and middle assemblies had an average $nI$ of only 0.048 and 0.061 (Fig 5A). Although *SCC* was not that powerful by itself, we found that a given value of 0-indegree led to a higher assembly membership probability if the innervation was significantly clustered (Fig 5B, first). This lead us to explore the correlation between 0-indegree and

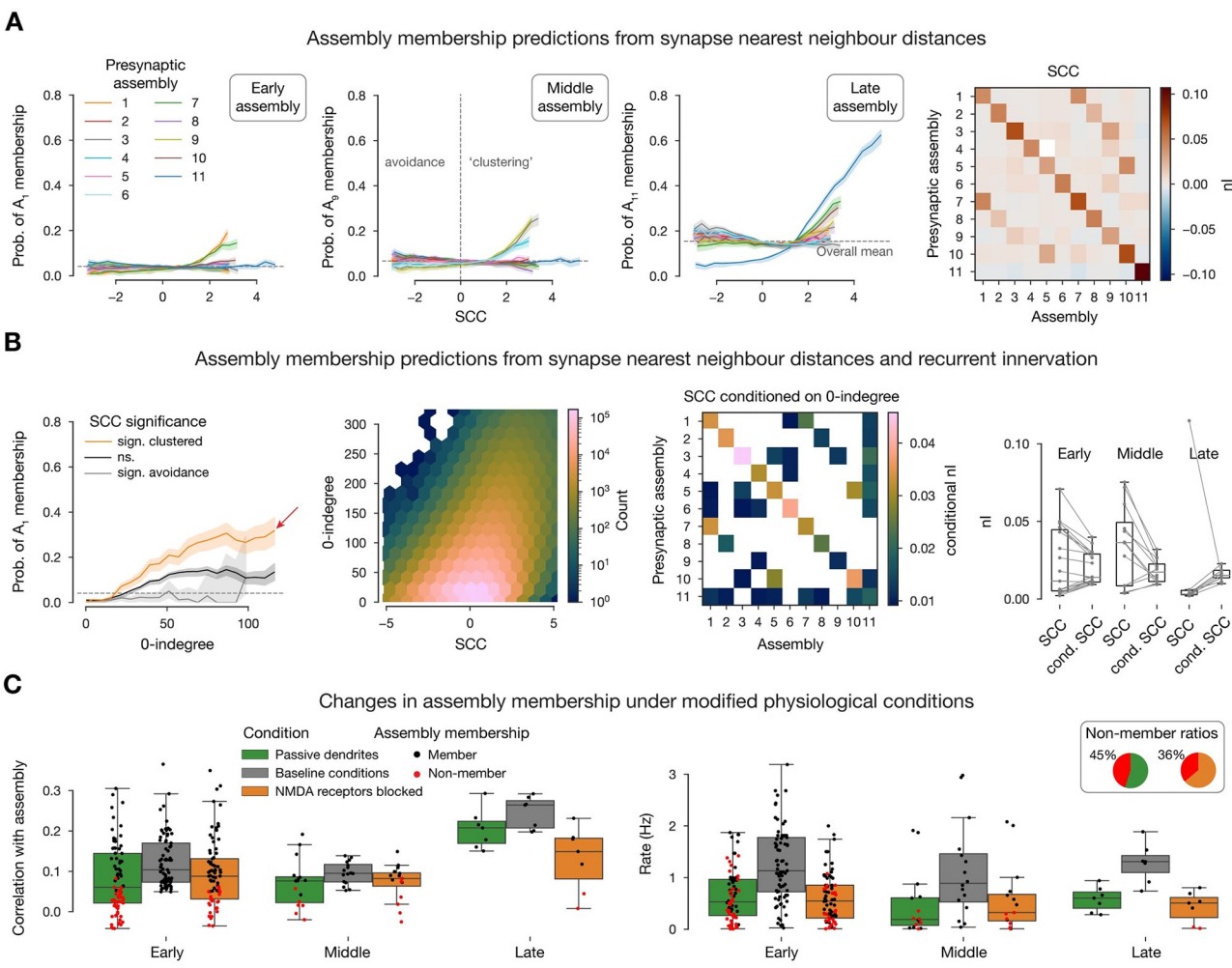

**Fig 5. Synapse clustering coefficient determines cell assembly membership. A**: Probability of membership in exemplary early (first), middle (second), and late (third) assemblies against synapse clustering coefficient (*SCC*, see Methods) with respect to all assemblies. Fourth: *nI* (normalized mutual information, see Methods) of *SCC* and assembly membership. **B**: Combined effect of *SCC* and indegree (as in Fig 4d). First: Probability of membership in an exemplary early assembly, against 0-indegree with respect to the same assembly, grouped by *SCC* significance (see Methods). Second: Joint distribution of *SCC* and 0-indegree. Third: *nI* of *SCC* and assembly membership conditioned on 0-indegree. Fourth: Relation of *nI* and conditional *nI* grouped by postsynaptic early, middle, and late assemblies (i.e., rows of the matrices in A fourth, and B third). **C**: Simulation results for 10 selected neurons per assembly (with the highest 0-indegree and significant clustering; red arrow in B left) with modified physiological conditions. Left: correlation of spike times with the assembly. Right: Single cell firing rates.

the *SCC* (Fig 5B second), finding a weak but significant correlation between these measures. Note that the *SCC* controls for the decrease in distance between synapses expected from a higher 0-indegree (see Methods), therefore we conclude that this is non-trival feature of the model. However, this means that the effects of 0-indegree and *SCC* on assembly membership are partially redundant. To dissociate these effects, we compute the *nI* between *SCC* and assembly membership conditioned by 0-indegree (see Methods), which is on average 0.025, reaching values up to 0.045 (Fig 5B third and fourth). This shows that 0-indegree and *SCC* affect assembly membership both independently but more so in conjunction.

The observed effect of *SCC* can only be explained by nonlinear dendritic integration of synaptic inputs [27, 28, 50]. Specifically, our model has two sources of nonlinearity: First, active ion channels on the dendrites [27, 49, 50], causing $Na^+$ and $Ca^{2+}$ spikes. Second, NMDA

conductances, which open in a voltage dependent manner, leading to NMDA-spikes [27, 50]. To show that these are the mechanisms by which the *SCC* acts, we studied how the removal of these non-linearities affects assembly membership of selected neurons. From each assembly we chose 10 neurons whose probability of membership was most affected by their *SCC*, i.e., those with highest 0-indegree combined with significant clustering (red arrow in Fig 5B first). We subjected these neurons to the same input patterns they received in the simulations (see Methods), but with passive dendrites (Fig 5C green) or blocked NMDA receptors (Fig 5C orange). These modifications altered the neurons' spiking activity, causing a non-negligible portion of them to drop out of their assembly, as their activity was no longer significantly correlated with it (Fig 5C first, see Methods). The manipulations resulted in a 45% drop in assembly membership for passive dendrites and 36% for blocking NMDA channels. Although, the manipulation resulted in an overall reduction of firing rate (Fig 5C second), this did not explain the drop in assembly membership (Fig 5C second: red dots with higher rate then black ones). We conclude that these nonlinearities contribute to the synchronization of activity within assemblies, underlying the observed effect of the *SCC*.

## Assemblies are robust across simulation instances

The results of our simulations are stochastic [51], leading to different outcomes for repetitions of the same experiment, as in biology. To assess the robustness of our results, we repeated our *in silico* experiment 10 times with the same thalamic inputs but different random seeds. Changing the seed mostly affects the stochastic release of synaptic vesicles [29, 51] especially at the low, *in vivo*-like extracellular $Ca^{2+}$ concentration used. The assemblies detected in the repetitions were similar to the ones described so far, in term of member neurons (Fig 6A),

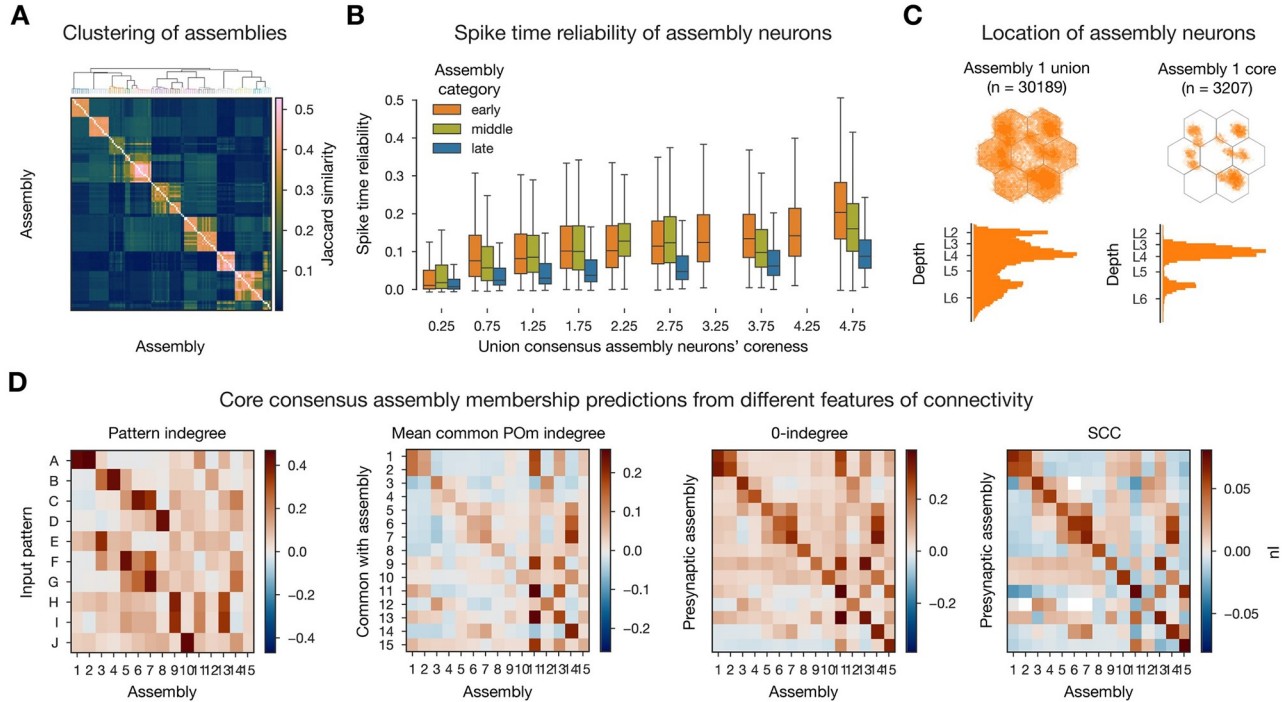

**Fig 6. Consensus assemblies. A**: Jaccard similarity based hierarchical clustering of assemblies from different simulation instances. **B**: Spike time reliability (see Methods) of neurons at different levels of coreness (see Methods). Missing boxes indicate missing categorical data at given levels of coreness. **C**: Locations of neurons in the union (all instances) and core (above expected, see Methods) of exemplary (pattern A responsive) early consensus assembly. **D**:*nI* (normalized mutual information, see Methods) of connectivity features and consensus assemblies membership.

temporal structure (early and middle groups and a non-specific late one, not shown), and their determination by connectivity features (not shown).

We hypothesize that cell assemblies in cortical circuits are inherently stochastic objects, partially determined by the structural connectivity, input stimuli, and neuronal composition. Thus, each repetition yields a different (but overlapping) set of assemblies and neurons contained in them. In order to get an approximation of these stochastic objects we pooled the assemblies detected in all repetitions and determined which best corresponded to each other by clustering them based on the Jaccard distance of their constituent neurons (see Methods, Fig 6A). According to this distance, nearby assemblies have a large intersection relative to their size. We called the resulting clusters *consensus assemblies*, and the assemblies contained in each its *instances*. We assigned to neurons different degrees of membership in a consensus assembly, based on the fraction of instances they were part of, normalized by a random control, and called it its *coreness* (see Methods). We found, that as coreness increased so did the neurons' spike time reliability (defined across the repetitions of the experiment [51, 52], see Methods), especially for the thalamus driven early assemblies (Fig 6B). We call the *union consensus assembly* the union of all its instances; and the *core consensus assembly* the set of cells whose coreness is significantly higher than expected (see Methods, see union vs. core distinction of Fig 6C).

When repeating our structural analysis on the core consensus assemblies, we found higher values of *nI* than before in all cases, except for the *SCC* (Figs 6D and 7). This indicates that assembly membership in this model is not a binary property but exists on a spectrum from a highly reliable and structurally determined core to a more loosely associated and less connected periphery. The notion of consensus assemblies is a way of accessing this spectrum.

Another way to take all 10 repetitions into account would be to average the time-binned spike trains of the simulated neurons. Thus, we averaged the input instead of the output of the assembly detection pipeline and we call these the *average assemblies*. We first compared these

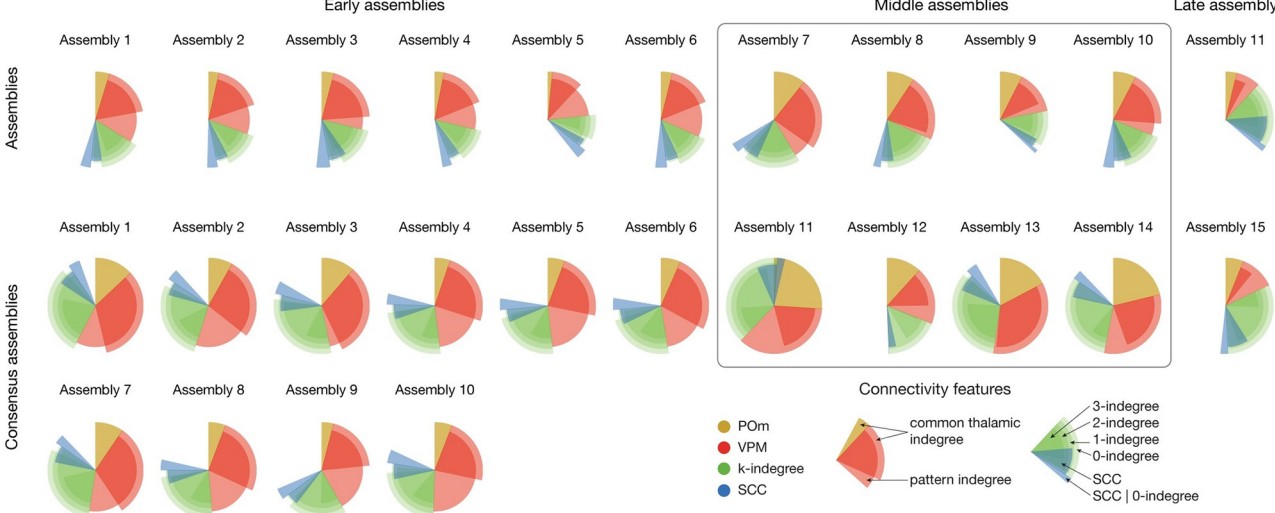

**Fig 7. Summary of nIs of all connectivity features and assembly membership.** Assemblies from single simulation on top and consensus assemblies from 10 simulations on the bottom. Only within-assembly interactions (diagonals of *nI* matrices) shown, except for the patterns, where for each column (postsynaptic assembly) we used the maximum value (strongest innervating pattern). Not only the colors, but the radii of the pies code for features: Red (VPM) longer one depicts common-innervation (as yellow for POm) while the shorter one direct innervation from patterns. Green: Simplex dimension increases as length decreases (longest: 0-indegree, shortest: 3-indegree). Blue: longer one is the *SCC* conditioned on 0-indegree, while the shorter one is the same, but unconditioned (and that is why it always overlaps with 0-indegree).

to the assemblies obtained in a single repetition. The similarity of significant time bins was higher for average assemblies (S6B1 Fig), and their sizes were larger (S6D Fig second panel), with neurons belonging to up to 10 assemblies out of the 13 detected (S6D Fig third panel). On the other hand, the $nI$ of the structural features and membership remained the same as for a single repetition (compare matrices in Figs 4 and 5 to S6C Fig).

When contrasting the average and consensus assemblies, we found pairs with high Jaccard similarity (S6D and S6E1 Fig). A detailed comparison showed that all neurons that are part of at least 6 out of 10 assembly instances were all contained in their matching average assembly. Further lowering the cutoff began admitting neurons that were not part of the corresponding average assembly (S6E2 Fig). On the other hand, there are barely any cells in the average that are not contained in the union consensus assembly. This demonstrates how assembly membership becomes less determined towards the periphery, mirroring the reduced $nI$ with the structural connectivity features. Furthermore, a neuron in a consensus assembly will most likely belong to all instances (S6E2 Fig), unlike for the binomial distribution expected by chance.

In summary, while average assemblies give similar results to the union consensus assemblies, the coreness values used in the consensus assembly framework assign different degrees of membership to its neurons that can be taken into account in downstream analyses, e.g., by considering only the functionally reliable core. Furthermore, for this core the determination by most structural features, measured by $nI$, is stronger (Fig 7). Note in particular, that structural properties do not fully explain neuron membership in an assembly. This is to be expected since non-linear neural network dynamics and their connection to their underlying structure are notoriously difficult to describe [53, 54]. This is in part due to the rich variety of dynamical states observed even in networks with more homogeneous connectivity such as uniform connectivity or distance dependent connectivity [16, 19, 55, 56].

## Discussion

Using a detailed, large-scale cortical network model we examined the link between cortical structure and function. Our principal findings from analyzing the connectivity of functional cell assemblies are as follows: (1) Different afferents dominated determination of membership in assemblies linked to different time windows: VPM innervation affected membership in early assemblies, POm innervation in middle assemblies, while recurrent innervation in the late assembly (Fig 7 red, yellow, green). (2) Recurrent innervation more strongly facilitated assembly membership the more the innervating neurons were wired among themselves, adding a non-local component to the structure-function relation. (3) Similarly, the innervation of a neuron by an assembly was significantly more powerful in facilitating membership when its synapses were clustered on the dendrites (Fig 7 blue). (4) In conjunction with the structure of cortical connectivity, features of subcellular physiology such as active dendritic channels and NMDA receptors, also influence assembly membership. (5) Interactions between assemblies emerged, where innervation by one assembly explained membership in another. Positive interactions (with increased membership probability) were found when the direction of innervation reflected the temporal order of assembly activation; otherwise, weaker or even negative interactions were found.

Point (1) above confirms our previous findings that, while the presence of an external stimulus makes the circuit much more reliable, this effect is not merely driven by direct innervation, but also requires recurrent connectivity [51]. Point (2) predicts a functional consequence for non-random features of neuronal connectivity, such as nodes with high centrality values and the presence of a rich club, that have been characterized in many species and regions, at

various levels of resolution [57]. Points (3) and (4) link the theory of neuronal assemblies to the literature on active dendritic computation [26–28]. Point (5) strengthens the case that the topology of connectivity is best studied in a directed way, since undirected networks (although more amenable to network science methods) miss an essential part of the picture. In particular, the relationship between the structural directionality of the connection and the temporal direction of the flow of activity [25]. Our principal findings on the structural features shaping assembly membership are robust with respect to the choice of threshold during hierarchical clustering and the time bin size parameters (S2 and S7 Figs respectively). While changes in these parameters lead to a different number of assemblies, the *nI* values of the underlying structural features remained in the same range (S7C Fig). Additionally, when the number of clusters was fixed to be the same across different time bin sizes we found a strong one-to-one correspondence between the member neurons of the resulting assemblies and those detected for 20 ms time bins. (S7D2 Fig).

We have shown that assembly membership is determined by certain structural prerequisites, mostly amounting to increased membership probability when more afferent synapses from various sources are formed on a neuron. These may be costly to fulfill, both in terms of energy [58] and space taken up by wiring. Chklovskii et al. [59] considered efficient layouts of wiring, i.e., axons and dendrites, given a certain connectivity matrix. They found that the layout is tightly constrained by the available space and close to a theoretical optimum. Here, we expanded on that idea, demonstrating that on top of it, the structure of the connectivity matrix is efficient. Indeed, on the sub-cellular level, we have shown, that synaptic clustering on dendrites increases efficiency, in that the same probability of assembly membership can be obtained with about 50% of the indegree when the synapses are highly clustered (Fig 5B, S8A Fig). Furthermore, on the connectivity level, another potential mechanism lies in the non-local interaction of the presynaptic population measured by the *k*-indegree. We have shown that for higher values of *k*, a given membership probability is attained for lower *k*-indegree values, suggesting higher efficiency (Fig 4E, left). However, as *k* grows, *k*-indegree counts the innervation by larger and larger motifs, potentially requiring more incoming synaptic connections (S8B2 Fig). Here, we control for this by repeating our analysis not with respect to the number of motifs but the size of the presynaptic subpopulation forming them. When going up from $k = 0$ to $k = 1$, $k = 2$, and $k = 3$ the same membership probability can be obtained for a given neuron with about 80%, 34% and 6% of the incoming connections (S8B1 Fig), confirming that non-local interactions in the presynaptic population make afferent innervation more efficient. This can be explained by our earlier finding that the simplicial motifs we considered increase the correlations and reliability of the spiking activity of participating neurons [25].

Our predictions listed above are experimentally testable *in theory*, and given the state of experimental techniques may soon be testable *in practice*. Required data for our analyses is densely reconstructed connectivity with co-registered neuron activity. The MICrONS $mm^3$ dataset [60] combines connectivity at subcellular resolution with neuronal activity data. However, we found that the set of neurons with available co-registered activity was too sparse in that dataset: When we randomly sampled neurons with the same density in our model, the characterized trends were not visible (not shown). Yet, the accelerated pace of the evolution of activity recording techniques suggests that the required density of recordings to test this pipeline's predictions might be achieved in the near future.

In our simulations, we found a single, non-specific assembly being activated by all 10 stimuli. This is unexpected, as such a repeated activation would be energetically expensive, but non-informative. This can be explained by two hypotheses: First, the model represents a cortical microcircuit isolated from the rest of cortex. It is possible that *in vivo* late assemblies are determined in conjunction with top-down feedback from other cortical regions. Indeed, late

assemblies had lower firing rates of participating neurons (S1C Fig), indication of possibly missing extrinsic excitatory input. One top-down source in our model was the POm input, which we made non-stimulus-specific. Leaving out this non-specific component of the stimuli did not lead to a stimulus-specific late assembly (S9 Fig). This could be further studied in the future using simulations of coupled brain regions. Second, while our network model has strongly non-random connectivity, constrained by neuronal morphology [61, 62], many parameters are initialized randomly within those constraints that would *in vivo* be determined through plasticity mechanisms, such as maximal synaptic conductance and which potential synaptic contacts are active. As such, it should be viewed as a circuit in a non-random plastic state, but unshaped by experience. It is possible that specificity of late assemblies emerges through plasticity, while learning to distinguish relevant stimuli. The presence of assemblies at all in a naive circuit is in line with the belief that the brain is not a tabula rasa [63]. It is also in line with the recent *in vivo* experiments of Bathellier et al. [9] and Tragenap et al. [64], who found endogenous cell assemblies in mouse auditory and ferret visual cortices. Furthermore, Tragenap et al. [64] also found that these endogenous assemblies *solidify* and become more reliable after eye opening.

These points lead to the question: How does long-term plasticity affect the three aspects discussed above, i.e., assembly wiring efficiency, competition for member neurons and assembly solidification? Recent modelling studies investigated how plasticity rules and network activity lead to assembly formation, maintenance and *competition for member neurons*, but they started from non-structured initial connectivity without assemblies [20–22]. To provide a complementary view, our model and network simulations could be augmented with models of structural or functional plasticity. The analysis methods introduced here then provide new, quantitative ways to characterize assemblies, their temporal evolution and the connectivity underlying them.

## Methods

### Network simulations

The most recent version of the detailed, large-scale cortical microcircuit model of Markram et al. [29] was used for the *in silico* experiments in this study. Updates on its anatomy, e.g., atlas-based cell densities are described in Reimann et al. [37], while updates on its physiology e.g., improved single cell models and missing input compensation in Isbister et al. [32]. The 2.4 $mm^3$ subvolume of the juvenile rat somatosensory cortex, containing 211,712 neurons is freely available at: https://zenodo.org/record/7930275.

Although large-scale with bio-realistic counts of synapses originating from the local neurons, the neurons in the circuit still lacked most of their synapses (originating from other, non-modeled regions) [29]. In order to compensate for this missing input, layer and cell-type specific somatic conductances following an Ornstein-Uhlenbeck process were injected to the cell bodies of all neurons [65]. The noise sources were stochastically independent between neurons. The algorithm used to determine the mean and variance of the conductance needed to put the cells into an *in vivo*-like high-conductance state, and the network as a whole into an *in vivo*-like asynchronous firing regime with low rates and realistic responses to short whisker stimuli is described in Isbister et al. [32]. The *in vivo*-like state used in this article is the same as $[Ca^{2+}]_o$ = 1.05 mM, percentage of reference firing rates = 50%, CV of the noise process = 0.4 from Isbister et al. [32].

Simulations of selected cells with modified physiological conditions (active dendritic channels blocked or NMDA conductance blocked) used the activity replay paradigm of Nolte et al. [51]. In short, for each of these cells, spike times of its presynaptic population were recorded in

the original network simulation. Then the selected cells were simulated in isolation by activating their afferent synapses according to the recorded spike times in the network simulation. Thus, the modified activity of the isolated cell did not affect the rest of the network.

Simulations were run using the NEURON simulator as a core engine with the Blue Brain Project's collection of hoc and NMODL templates for parallel execution on supercomputers [66–68]. Simulating 2 minutes of biological time took 100,000 core hours, on our HPE based supercomputer, installed at CSCS, Lugano.

### Distance metrics

This section gives a brief overview and justification of the various distance metrics used below.

Population activity in time bins were compared using their cosine similarity [10]. More precisely, for any two time bins, we considered their firing rate vectors $\vec{t_1}, \vec{t_2} \in \{0, 1\}^N$ where $N$ is the number of neurons, describing the neurons firing in that time bin. They were then compared by

$$\text{cosine similarity}(\vec{t_1}, \vec{t_2}) = \frac{\vec{t_1} \cdot \vec{t_2}}{\|\vec{t_1}\| \|\vec{t_2}\|},$$

where $\cdot$ denotes the dot product and $\|\cdot\|$ is the Euclidean norm. Two time bins with high cosine similarity have similar sets of firing neurons, thus detecting co-firing. Note that there is an increasing relationship between firing rate and cosine similarity ([69], S1A2 Fig).

Input patterns were defined using the Hamming distance between the sets of VPM fiber bundles involved to have specific values and thus specific sizes of intersections (Fig 2B). More precisely, for any two patterns, let $\vec{P_1} \ \vec{P_2}$ be the binary vectors of VPM fiber bundles involved in them and $N_F$ the total number of fibers. Their Hamming distance is

$$N_F - \vec{P_1} \cdot \vec{P_2}.$$

Input patterns were compared using Earth mover distances between the flat map locations of their contained fibers (Fig 3E). This distance indicates the minimal amount of work required to transform one pattern into another by interchanging/moving the activated fibers; see Rubner et al. [70] for a detailed definition.

Assemblies of neurons were compared using their Jaccard distances. Like Hamming, this also compares sizes of intersections but it is normalized with respect to the sizes of the sets involved. That is, for any two assemblies, let $\vec{A_1} \ \vec{A_2}$ be the binary vectors of neurons involved in them and $N_1, N_2$ the number of neurons in each. Their Jaccard distance is

$$\frac{N_1 + N_2 - \vec{A_1} \cdot \vec{A_2}}{N_1 + N_2}.$$

Assembly sequences, i.e., vectors of size the number of assemblies, with each entry counting the number of time bins the corresponding assembly was active in response to an input pattern, were compared using their normalized Euclidean distances. This is, for any two such vectors, say $\vec{v_1}, \vec{v_2}$,

$$d\left(\frac{\vec{v_1}}{\|\vec{v_1}\|}, \frac{\vec{v_2}}{\|\vec{v_2}\|}\right),$$

was computed, where d denotes Euclidean distance.

Finally, afferent synapses were compared using their path distances. Specifically, the dendritic tree was represented as a graph with nodes being its branching points and edges between

them weighted according to the length of sections connecting them. Distances between synapses was computed as the path distance in this graph. That is, between any two branching points represented by nodes $u, v$

$$\min_{p:u \to v} \{l(p)\},$$

was computed, where the minimization is done over all paths $p$ from $u$ to $v$ in the graph and $l(p)$ denotes its length, i.e., number of edges in $p$.

## Thalamic input stimuli

For thalamic input stimuli modeled afferent fibers from two thalamic nuclei, VPM and POm were used. The VPM input spike trains were similar to the ones used in Reimann et al. [35]. In detail, the 5388 VPM fibers innervating the simulated volume were first restricted to be $\leq 500$ $\mu$m from the middle of the circuit in the horizontal plane to avoid boundary artifacts. To measure these distances we use a flat map, i.e., a two dimensional projection of the volume onto the horizontal plane, orthogonal to layer boundaries [37, 71]. Second, the flat map locations of the resulting 3017 fibers were clustered using k-means to form 100 bundles of fibers. The base patterns (A, B, C, and D) were formed by randomly selecting four non-overlapping groups of bundles, each containing 12% of them (corresponding to 366 fibers each). The remaining 6 patterns were derived from these base patterns with various degrees of overlap (see beginning of Results, Fig 2B). Third, the input stream was defined as a random presentation of these ten patterns with 500 ms inter-stimulus intervals, such that in every 30 second time intervals every pattern was presented exactly six times. Last, for each pattern presentation, unique spike times were generated for its corresponding fibers following an inhomogeneous adapting Markov process [72]. When a pattern was presented, the rate of its fibers jumped to 30 Hz and decayed to 1 Hz over 100 ms. For the calibration of the maximum firing rates, the rate of VPM fibers from Isbister et al. [32] was used a starting point, and then systematically lowered until the evoked network activity remained stable. This reduction of the rate was needed because of the long-lasting inputs used here, in contrast to the whisker flick-like, short (2 ms) VPM inputs in Isbister et al. [32]. Inputs from POm were non-specific, i.e., the same randomly selected 12% of the (unclustered) 3864 POm fibers were activated each time any pattern was presented (in every 500 ms). The spike trains were designed with the same temporal dynamics as described above for VPM (and were thus unique for all presentations), but with half the maximum rate (15 Hz). The implementation of spike time generation was based on Elephant [73].

## Assembly detection

Our assembly detection pipeline was a mix of established techniques and consisted of five steps: binning of spike trains, selecting significant time bins, clustering of significant times bins via the cosine similarity of their activity, and determination of neurons corresponding to a time bin cluster and thus forming an assembly. Note that time bins instead of neurons were clustered because this allows neurons to belong to several assemblies.

Spikes of excitatory cells were first binned using 20 ms time bins, based on Harris et al. [5], who take a postsynaptic reader neuron specific point of view. They suggest 10–30 ms as an ideal integration time window of the presynaptic (assembly) spikes [3, 5]. Next, time bins with a significantly high level of activity were detected. The significance threshold was determined as the mean activity level plus the 95th percentile of the standard deviation of shuffled controls. The 100 random controls were rather strict, i.e., all spikes were shifted only by one time bin forward or backward [7, 10]. Next, a similarity matrix of significant time bins was built, based

on the cosine similarity of activation vectors, i.e., vectors of spike counts of all neurons in the given time bins [10]. The similarity matrix of significant time bins was then hierarchically clustered using Ward's linkage [12, 13]; a method that minimizes the total within-cluster variance. Potential number of clusters were scanned between five and twenty, and the one with the lowest Davis-Bouldin index was chosen, which maximizes the similarity within elements of the cluster while minimizing the between-cluster similarity [40]. Alternatively, a given number of clusters can be directly chosen by the user. These clusters corresponded to potential assemblies.

As the last step, neurons were associated to these clusters based on their spiking activity, and it was determined whether they formed a cell assembly or not. In detail, the correlations between the spike trains of all neurons and the activation sequences of all clusters were computed and the ones with significant correlation selected. Significance was determined based on exceeding the 95th percentile of correlations of shuffled controls (1000 controls with spikes of individual cells shifted by any amount [12, 30]). In relation to Fig 5C left, it is important to note, that these correlation thresholds were specific to a pair of a neuron and an assembly. Finally, it is possible to have a group of neurons that is highly correlated with one part of the significant time bins in a cluster, and another that is highly correlated with the rest, while the two groups of neurons have uncorrelated activity. To filter out this scenario, it was required that the mean pairwise correlation of the spikes of the neurons with significant correlations was higher than the mean pairwise correlation of neurons in the whole dataset [30]. Clusters passing this test were considered to be functional assemblies and the neurons with significant correlations their constituent cells.

For a test of the methods on synthetic data and comparison with other ways of detecting cell assemblies please consult Herzog et al. [30]. Assembly detection was implemented in Python and is publicly available as `assemblyfire`.

## Calculation of information theoretical measurements

To quantify the structural predictability of assemblies, the *mutual information* of assembly membership ($Y_n$) and a structural feature of a neuron ($X_m$) was used, which is a measure of the mutual dependence between the two variables [74]. More precisely, $Y_n$ is a binary random variable that takes the value 1 if a neuron belongs to an assembly $A_n$, and 0 otherwise. On the other hand, $X_m$ is a random variable determined by a structural property of the neuron with respect to the assembly $A_m$, e.g., the number of afferent connections into the neuron from all neurons in $A_m$.

To assess the dependence between these two variables, first the dependence of the probability of a neuron belonging to assembly $A_n$ given a specific value of the structural feature measured by $X_m$ was studied. More precisely, the function $f_{n,m}$ whose domain is the values of $X_m$ and is given by:

$$f_{n,m}(x) = P(Y_n = 1 | X_m = x)$$

was considered. If this function has an increasing or decreasing trend, then the random variables $Y_n$ and $X_m$ can not be independent.

Their dependence was quantified by means of their mutual information. The value of mutual information is always non-negative and it is zero when the random variables are independent. In order to restrict this value to [0, 1] it was divided by the entropy of $Y_n$, which measures the level of inherent uncertainty of the possible outcome of the values of $Y_n$ [75]. The calculation of mutual information is based on the probabilities of $X_m$ and $Y_n$ across all possible outcomes. If the number of possible values of $X_m$ is large compared to the number of samples,

there can be errors in determining these probabilities, possibly leading to inflated values of the mutual information. Therefore, the values of $X_m$ were binned into 21 bins between the 1st and 99th percentile of all sampled values. The number of bins was determined such that the resulting value of mutual information in a shuffled control did not exceed 0.01. Shuffled controls (one per pair) were also used to threshold the mutual information values by considering only the pairs $(n, m)$ whose mutual information was larger than the mean plus one standard deviation of all pairs in the shuffled controls. Finally, a negative sign was added to the significant mutual information value if the function $f_{n,m}$ was decreasing i.e., the probability of membership in $A_n$ decreased as the values of $X_m$ increased. This was assessed by the slope of a weighted (by the number of samples in each bins) linear fit of the function $f_{n,m}$. This normalized, thresholded and signed mutual information value was called *normalized mutual information* and denoted $nI(Y_n, X_m)$.

All the statements above can be made conditional with respect to a third random variable, yielding the *conditional normalized mutual information*, which was used when two structural features were inherently believed to be interacting as in the case of the *SCC* and indegree.

Calculations were done with the `pyitlib` package.

## Synaptic clustering coefficient

To quantify the co-localization of synapses on the dendrites of a neuron $i$ from its presynaptic population $P_i$ with a single, parameter-free metric, synaptic clustering coefficient *SCC* was defined and calculated for all excitatory neurons in the circuit with respect to all assemblies. Based on these locations, $D_i$, the matrix of all pairwise path distances between synapses on $i$ from $P_i$ were calculated. Let $D_{i,p}$ be the submatrix for pairs of synapses originating from a subpopulation $p \subseteq P_i$. Then the nearest neighbour distance for $p$ can be written as:

$$nnd(i, p) = \text{mean}\left(\min_{rows}(D_{i,p})\right)$$

In particular, for the subpopulation of $P_i$ of neurons in the assembly $A_n$, denoted by $p_n$, $nnd(i, A_n) = nnd(i, p_n)$ was defined, where $p_n = P_i \cap A_n$. This value was normalized, using the *nnd* values of 20 random presynaptic populations from $P_i$ of the same size as $p_n$. In summary, the *SCC* was defined as the negative z-score of $nnd(i, A_n)$ with respect to the distribution of control *nnd*s (S5 Fig). Additionally, the significance of the clustering or avoidance of the synapse locations was determined with a two-tailed t-test of $nnd(i, A_n)$ against the 20 random samples with an alpha level of 0.05. *SCC* was implemented using `NeuroM` and `ConnectomeUtilities`.

## Determination of consensus assemblies

Consensus assemblies were defined over multiple repetitions of the same input stream. These were groups of assemblies with similar sets of neurons. Additionally, all assemblies in a group were required to originate from a different repetition; noted as the *repetition separation criterion*. The Jaccard distance matrix between all pairs of assemblies from all repetitions were computed and modified by setting the distances between pairs of assemblies from the same repetition to twice the maximum of the whole matrix. The matrix was then hierarchically clustered using Ward's linkage, and the lowest number of clusters that satisfied the repetition separation criterion was chosen. The resulting clusters were the *consensus assemblies*, and the assemblies within them their *instances*.

The *union consensus assembly* was defined as the set of neurons given by the union of all instances. Its member neurons were assigned a membership degree based on number of instances they were part of in two ways. First, by simply using the fraction of the instances a

neuron was part of. Second, in what was called the *coreness* value of a neuron, which is the number of instances a neuron is part of normalized by its expected value, given the number and sizes of its instances. This was calculated as a binomial distribution with $n$ set to the number of instances and $p$ to the mean size of the instances, divided by the size of the union consensus assembly. Based on this, the coreness of a neuron contained in $r$ instances was defined as $-log_{10}(1 - B_{n,p}(r))$, where $B$ is the cumulative binomial distribution. Neurons with a coreness value exceeding 4 were considered to be part of the corresponding *core consensus assembly*.

### Calculation of spike time reliability

Spike time reliability was defined as the mean of the cosine similarities of a given neuron's mean centered, smoothed spike times across all pairs of repetitions [52, 69]. To smooth the spike times, they were first binned to 1 ms time bins, and then convolved with a Gaussian kernel with a standard deviation of 10 ms.

### Supporting information

**S1 Fig. Spikes of (significant) time bins. A1** Cosine similarity matrix (same as in Fig 3B1, but unsorted). **A2**: Joint distribution of pair-wise mean firing rate (within significant time bins) and cosine similarity. **B1**: 2D linear projection of mean centered and normalized spike matrix. **B2**: 2D nonlinear projection of spike matrix (using cosine distance). **C**: Single cell firing rates of neurons belonging to different assemblies. **D**: Distribution of stimulus evoked cluster sequence lengths (rows of matrices in Fig 3B2) and number of spikes in these significant time bins.
(TIF)

**S2 Fig. Different number of clusters (of significant time bins) tested. A1, B** and **C**: As in Fig 3B and 3E but for different number of clusters ($n$). **A2**: Davis-Bouldin index [40] for different number of clusters.
(TIF)

**S3 Fig. Cell assembly detection in L2/3. A-C**: as in Fig 3B, 3D and 3E. **d**: Jaccard similarity of assemblies (detected across layers, but restricted to L2/3 here) and the ones detected in L2/3 on the left, and number and location: flatmap view on top, depth-profile below of exemplary pairs (pattern A and B responsive ones) of assemblies with high similarity to its right.
(TIF)

**S4 Fig. Anatomy of thalamocortical synapses. A**: Density profile of VPM and POm synapses, digitized from Meyer et al. [41]. **B**: Synapse-to-soma path distances of different thalamocortical synapses on L5 neurons in an exemplary middle assembly ($A_8$). Box widths represent the ratio of the number of synapses (e.g. most synapses on $A_8$ L5 pyramidal cells are coming from pattern G, in line with the indegree based $A_8$ membership probability on Fig 4B second).
(TIF)

**S5 Fig. Synaptic clustering coefficient. A1**: Exemplary L5 pyramidal cell and all its afferent synapses from $A_{11}$ (in blue) and from a control group (one out of the twenty) with the same number of presynaptic neurons (in gray). **A2**: Zoom in on A1. Soma, basal dendrites, and proximal apical dendrites are visible. Axon is not shown. The rendering was done with the `BioExplorer` package. **B1**: Distance matrix between all pairs of $A_{11}$ synapses and distribution of nearest neighbour distances (minimum over the rows of the matrix) on its right (see Methods). **B2**: same as B1 but for the control group. The equations on the righmost part of the

figure are motivated and explained in the Methods.
(TIF)

**S6 Fig. Assemblies detected from averaged spike matrices. A, B**: as in Fig 3A and 3B. **C**: as in Fig 6C. **D**: Left: Jaccard similarity of consensus assemblies and average assemblies. Middle: Number of neurons in conesensus assemblies' union and core and average assemblies. Right: As middle, but number of neurons participating in given number of assemblies. **E1**: Jaccard similarity of consensus assemblies at different fraction thresholds (consensus assembly size grows to the right) and average assemblies. **E2**: Detailed comparison of the pair with the highest similarity in e1 at given number of assembly instances contained. (10 means that the consensus assembly neuron is part of 10/10 assembly instances, thus consensus assembly sizes grows to the right again.) Average \consensus is negligible (419 neurons) and is not shown.
(TIF)

**S7 Fig. Cell assembly detection with different time bins. A-B**: As in Fig 3B, but with different time bins. A: 10 ms, B: 40 ms (original: 20 20ms). **C**: As in Fig 4B and 4D. **D1**: Jaccard similiarity of assemblies detected this way and the original ones. **D2**: As D1, but the clustering tree cut at a different location (see S2 Fig) to result in 11 assemblies in the new cases as well.
(TIF)

**S8 Fig. Efficiency of innervation.** Probabilities of within assembly memberships for all assemblies. **A**: As in Fig 5B left. **B1**: Similar to Fig 4E left, but with different x-axis (presynaptic population size instead of simplex counts, see B2). **B2**: Illustration of the difference between $k$-indegree and presynaptic population size.
(TIF)

**S9 Fig. Cell assembly detection without POm input. A-B** As in Fig 3A and 3B, but the underlying simulation in this case does not have input from POm fibers.
(TIF)

## Acknowledgments

The authors thank Nicolas Ninin for his involvement in the early stage of this project, Elvis Boci and Cyrille Favreau for their help with visualizations and Alberto Antonietti, Christoph Pokorny, Kathryn B. Hess, Ran Levi, and Henry Markram for discussions.

## Author Contributions

**Conceptualization:** András Ecker, Daniela Egas Santander, Michael W. Reimann.

**Data curation:** András Ecker, Sirio Bolaños-Puchet, James B. Isbister.

**Formal analysis:** András Ecker, Daniela Egas Santander, Michael W. Reimann.

**Investigation:** András Ecker.

**Methodology:** András Ecker, Daniela Egas Santander, Michael W. Reimann.

**Project administration:** Michael W. Reimann.

**Software:** András Ecker, Daniela Egas Santander, Sirio Bolaños-Puchet, James B. Isbister, Michael W. Reimann.

**Supervision:** Michael W. Reimann.

**Validation:** András Ecker, Daniela Egas Santander, Sirio Bolaños-Puchet, James B. Isbister, Michael W. Reimann.

**Visualization:** András Ecker.

**Writing – original draft:** András Ecker, Daniela Egas Santander, Michael W. Reimann.

**Writing – review & editing:** András Ecker, Daniela Egas Santander, Sirio Bolaños-Puchet, James B. Isbister, Michael W. Reimann.

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
