## [Decision Letter · Decision Letter 0]

14 Jan 2024

Dear Mr. Ecker,

Thank you very much for submitting your manuscript "Cortical cell assemblies and their underlying connectivity: an in silico study" for consideration at PLOS Computational Biology. As with all papers reviewed by the journal, your manuscript was reviewed by members of the editorial board and by several independent reviewers. The reviewers appreciated the attention to an important topic. Based on the reviews, we are likely to accept this manuscript for publication, providing that you modify the manuscript according to the review recommendations.

In your revised manuscript, please make sure to take into account all the points raised by the reviewers, in particular those on spontaneous activity and the non-specific input. Please, also make sure to add line numbers, as required by reviewer 1. 

Sincerely,

Hugues Berry

Academic Editor

PLOS Computational Biology

Daniele Marinazzo

Section Editor

PLOS Computational Biology

Reviewer's Responses to Questions

**Comments to the Authors:**

Reviewer #1: # Summary and general comments

In this manuscript, Ecker, Santander et al conduct an in-silico study of a state-of-the art, biophysically detailed simulation of seven cortical columns of rat's primary somatosensory cortex. This model has been described and characterized in separate preprints (refs [35] and [36] in the text), as the reduced version (~200K neurons) of a larger model (4.2M neurons). The study is centered on the characterization of assembly-like, stimulus-evoked activity, and on the relation between assembly membership and circuit's anatomical structure. I find the work well written, original, technically sound, and quite remarkable in terms of biophysical realism. However it appears to me that most of the results are the direct outcome of the circuit's structural and dynamical properties. In summary, repeated patterns of activity are injected through thalamic afferents, activating consistently specific sets of neurons (early assemblies), delayed inputs facilitate activations in other sets of neurons (mid assemblies), finally recurrent connectivity causes further signal propagation across the whole population (late assembly). Factors such as calcium spikes and nonlinear interactions between clustered synapses also favor neural activations, when present, increasing transmission efficiency.

I find this work an interesting example of how a bottom-up approach can consistently and cohesively reveal large-scale circuit properties, offering more quantitative insights and predictions precluded to simpler models of pointwise neurons. Especially in quantifying more precisely how different circuit properties concur in generating evoked assembly-like activity. There are however some noteworthy limitations, that is important to mention.

First and foremost, the work focuses exclusively on assemblies directly evoked by injected stimuli, with no mentions of spontaneous activity. However a lot of interest is devoted to why and by which mechanisms assemblies emerge during spontaneous activity, and on the similarities between the spontaneous assemblies and those evoked by stimuli (e.g. refs [10,11,13,58] in main text). I think it should be explained why this aspect has not been investigated in the paper, despite its centrality in both experimental and modeling work.

Second, I find it curious that about 150 ms after stimulus onset the same, broadly distributed late assembly activates for every stimulus in the input set. This is an interesting model prediction I am not fully convinced about (it seems unjustified in terms of energy cost, for a patterns that seems quite uninformative). I also wonder how much of it is due to the presence of the broad, unspecific POm component that comes with every input stimulus. For example it would be interesting to see if it is still present in the absence of POm stimulus, and to what extent. Perhaps it should be better explained and justified why input stimuli have both a clustered and a more spread-out component.

# Minor points

P0 - The manuscript does not have line numbers. I am surprised they are not a strict requirement, but even so please do add them in the future, as they would simplify the process of revising and editing.

Abstract - I find the mention of plasticity in the concluding sentence a bit confusing, since it is completely out of the scope of the work presented, and barely mentioned in the discussion.

P2, 4th paragraph. Typo on second line and question mark missing at end.

P2, 5th paragraph. "tens of thousands" but it's more like 200K. Assemblies are described as "groups of neurons that fire together more than expected". Intuitively the definition makes sense to me, but I worry there might be a bit too much subjectivity in the meaning of "more than expected".

P3, last paragraph of section 1. "Assemblies active after 50 ms are determined by recurrent connectivity". But later on the result is that "recurrent connectivity explains late assemblies". From Fig 3 B2 I would associate 50 ms with mid assemblies. (might be useful to offer a time range for mid and late assemblies)

P3, section 2.1 . Although the input is better described in the method section, I think you should describe the POm input as well. Initially I erroneously thought that VPM and POm stimuli belonged to separate sessions, rather than being delivered at the same time.

Fig 3e I would show the y axis in the central panel as well.

P5, section 2.1.1 Since the aim is not to build an ideal decoder, I don't think it can be stated that stimuli can "be distinguished better" in an analysis restricted to the L2/3 layer. I understand there is a better, one-to-one correspondence between detected assemblies and injected stimuli, but that would also be feasible in a full model with an optimal decoder, or with different assembly-selection criteria.

P7, section 2.2 In the definition of normalized mutual information, I would describe more clearly the distinction between having positive value, being zero, or having a negative value.

P10 , end of 2.2.2 section, states that 0-indegree is equivalent to indegree. I think this should be clarified a bit earlier, given that the x-axis in Fig. 4D simply reads "indegree"

P14, "Our analysis support the idea that neuronal activity revolves around activation of assemblies". I find this statement a bit too strong: the analysis considers assemblies evoked in the model by specific stimuli chosen for this purpose. There is no comparison with alternative ideas and perspectives. The "idea that neuronal activity revolves around activation of assemblies" seems to me the very starting point of this work.

P15, Discussion. I find the sentence "we are integrating the plasticity model of Chindemi et al" a bit too specific. It's surely important to indicate plasticity as possible future direction, but the conclusions could be a bit broader in scope.

P17 , POm stimulus. I suggest to mention this in the results section as well. Also, it is not clear whether the POm fibers were randomly sampled only once for each VPM pattern and associated to it as frozen noise, or resampled independently at each stimulus presentation. The result section simply states "The stream consisted of repeated presentations of 10 different input patterns in random order", suggesting that they are sampled only once.

Reviewer #2: Assemblies of neurons firing together in the isocortex have been studied using in vivo recording methods, and their activity has been linked to different aspects of cortical coding. However, these in vivo methods lack information about structural connectivity, cell type identity, and subcellular synaptic specificity. This study uses a preexisting large, biophysically detailed model of rat somatosensory cortex to investigate the structural contributions to the formations of coactive neuronal assemblies. The authors quantify the effects of thalamic innervation, recurrent connectivity, subcellular targeting in determining membership of cell assemblies that occur at different times after stimulus presentation to the model. They find that the factors can be predictive, although the degree of contribution changes based on the time of assembly activity.

Overall, the study is quite interesting, well-designed, and generally well-presented. The use of a model where connectivity and neuronal identities are entirely known is a powerful approach to testing the influence of structural factors. This approach could also generate predictions for future in vivo experiments (especially those that are combined with post hoc connectivity measurements, like large-volume electron microscopy reconstruction). I only have a few suggestions to potentially improve the clarity of the manuscript or add information of interest to the community.

There are a few places where I felt there was a slight disconnect between the text and the figures. In the last paragraph of page 8, the authors discuss the notion of "k-indegree with respect to an assembly", and discuss results for 0-indegree before moving to 1-, 2-, and 3-indegree. The results for 0-indegree are presented in Figure 4D, but there is no specific mention of "0-indegree" (the authors just use the term "indegree" without a modifier), so it was unclear to me at first which results were specific to the 0-indegree analysis. I think it would be clearer if the "0-indegree" term was explicitly used in Fig. 4D. After this text, the authors then go on to say they will refer to 0-indegree as just "indegree" in later figures, but I think in Figure 4 it would better align with the text if "0-indegree" was explicitly used.

Also, for the results shown in Figure 6, the authors define a measure of "coreness" (discussed in the results and methods), but it seems as though this measure is not directly presented in Figure 6. Figure 6B has neurons grouped by "fraction of assemblies" instead, which seems to be similar to the "coreness" metric but not exactly the same. I think it would be clearer if a consistent measure were discussed and presented (or at least describe the differences between the measures).

Lastly, since cellular identity (as well as connectivity) is completely known in the model, I was curious if there were any cell-type specific results for assemblies or for the effect of innervating k-cliques. For example, are there k-cliques comprising particular combinations of cell types (types of inhibitory vs excitatory; types of excitatory cells within/across layers, etc.) that are more likely to lead to a cell joining an assembly than others? There is a fair amount of discussion of the layers in which the neurons belonging to different assemblies reside, but I think some additional analysis of cell type could be of great interest to readers interested in assigning circuit roles to particular types.

Overall, this is a strong and interesting study making good use of a large-scale, biophysically detailed model that should enable future directions of research.

Reviewer #3: see attachment

**Have the authors made all data and (if applicable) computational code underlying the findings in their manuscript fully available?**

Reviewer #1: Yes

Reviewer #2: Yes

Reviewer #3: Yes

PLOS authors have the option to publish the peer review history of their article (what does this mean?). If published, this will include your full peer review and any attached files.

Reviewer #1: No

Reviewer #2: No

Reviewer #3: No

Figure Files:

Data Requirements:

Reproducibility:

References:

---

## [Decision Letter · Decision Letter 1]

5 Feb 2024

Dear Mr. Ecker,

We are pleased to inform you that your manuscript 'Cortical cell assemblies and their underlying connectivity: an in silico study' has been provisionally accepted for publication in PLOS Computational Biology.

Best regards,

Hugues Berry

Academic Editor

PLOS Computational Biology

Daniele Marinazzo

Section Editor

PLOS Computational Biology

Reviewer's Responses to Questions

**Comments to the Authors:**

Reviewer #1: I thank the authors for further improving and clarifying on their very solid and detailed work. I am slightly dissatisfied seeing that the revised text did not elaborate much on why the model could not produce spontaneous activity resembling evoked assembly activations, and L73-74 appear to state the choice of studying only evoked assemblies rather than to motivate it. This said, I think the authors clearly indicated the scope of their work in the text, so I fully respect their choice of focusing on evoked dynamics.

I also appreciate the control experiment without the POm component, and the addition of interesting hypotheses on the late assembly shared across stimuli.

Some minor points:

L61 - I think a "?" is missing.

Figure S1 A2 - Please clarify the definition of pair-wise mean firing rate (of time bins).

L143,144 - To compare the layer-profiles of assemblies and thalamic input, I had to look at supplement Fig. S4 and compare it with 3C. It could be easier for the reader if an assembly layer profile was re-plotted in Fig S4.

Figure S2 - The caption should explain what is different from the previous figure, and in particular define "n" in S2C which does not appear anywhere else in the text.

L175 - I think the question should be re-laborated a little. In general, in-vitro and in-vivo measures are different since the first are computer-generated and the latter refer to specific living organisms. I would prefer words like "equivalent to" or "similar" or "reflect" rather than "same".

L203-206 - The text says that nI is defined only if the mutual information is statistically significant. However in the methods section nI seems to be always defined, as long as mutual information can be calculated. If there are "undefined" situations, then please specify for what conditions mutual information is deemed not statistically significant.

L443-339 - I find this part very interesting, but slightly confusing. The MICrONS dataset is described as "feasible", but the following sentence immediately contradicts the statement, explaining that the co-registered activity was too sparse. So in what sense the current experimental techniques are feasible?

L462 - A "plastic state" implies changes in synaptic efficacies and learning. However my understanding is that the model is, in fact, not in a plastic state?

L583-584 - Typo. A low Davis-Bouldin index maximizes the *difference* between (distinct) clusters, not their similarity.

Figure S4 - rogue ' characters in the caption. Also, I would say "box width" instead of "ratio of box widths"

L602 - "assemblyfire" does not seem publicly available. Please specify whether it is on some repository or released along with the paper.

Reviewer #2: In their revised manuscript, the authors have addressed my main concerns about the initial work. The manuscript has been clarified and strengthened. I only note a few points either introduced by the revisions or that remained from the first submission:

Minor points:

- Line 139: "...most neurons (in fact 40%) of neurons do not belong to any assembly" - Seems more accurate to say "a plurality of neurons" since 40% is not "most" but is still larger than any given assembly.

- Figures 4E, S8B1: Typo in title still there - "generalied" -> "generalized"

- Figure 5C: Typo in title - "physiologycal" -> "physiological"

- Figure 6B: Please add a note in the figure legend about why not every assembly category appears in every "coreness" bin (presumably because there were no neurons with that combination of assembly category & coreness value, but good to state explicitly so readers know it is not a graphical error)

Reviewer #3: The authors have addressed my questions and I found the paper is much clearer than before.

**Have the authors made all data and (if applicable) computational code underlying the findings in their manuscript fully available?**

Reviewer #1: Yes

Reviewer #2: Yes

Reviewer #3: Yes

PLOS authors have the option to publish the peer review history of their article (what does this mean?). If published, this will include your full peer review and any attached files.

Reviewer #1: No

Reviewer #2: No

Reviewer #3: No

---

## [Editor Report · Acceptance letter]

15 Feb 2024

PCOMPBIOL-D-23-01817R1 

Cortical cell assemblies and their underlying connectivity: an in silico study

Dear Dr Ecker,

I am pleased to inform you that your manuscript has been formally accepted for publication in PLOS Computational Biology. Your manuscript is now with our production department and you will be notified of the publication date in due course.

With kind regards,

Zsofi Zombor
